# A Rare Case of Malignant Ovarian Germ Cell Tumor: Dysgerminoma and Seminoma in the Same Patient

**DOI:** 10.3390/reports6010014

**Published:** 2023-03-03

**Authors:** Melinda-Ildiko Mitranovici, Diana Maria Chiorean, Sabin Gligore Turdean, Maria Cezara Mureșan, Corneliu-Florin Buicu, Raluca Moraru, Liviu Moraru, Titiana Cornelia Cotoi, Havva Serap Toru, Adrian Apostol, Claudiu Mărginean, Ion Petre, Ioan Emilian Oală, Viviana Ivan, Ovidiu Simion Cotoi

**Affiliations:** 1Department of Obstetrics and Gynecology, Emergency County Hospital Hunedoara, 14 Victoriei Street, 331057 Hunedoara, Romania; 2Department of Pathology, County Clinical Hospital of Targu Mures, 540072 Targu Mures, Romania; 3Department of Obstetrics and Gynecology, “Victor Babes” University of Medicine and Pharmacy, 2 Eftimie Murgu Sq., 300041 Timisoara, Romania; 4Public Health and Management Department, “George Emil Palade” University of Medicine, Pharmacy, Science, and Technology of Targu Mures, 540139 Targu Mures, Romania; 5Faculty of Medicine, “George Emil Palade” University of Medicine, Pharmacy, Sciences and Technology, 540142 Targu Mures, Romania; 6Department of Anatomy, “George Emil Palade” University of Medicine, Pharmacy, Sciences and Technology, 540142 Targu Mures, Romania; 7Department of Pharmaceutical Technology, “George Emil Palade” University of Medicine, Pharmacy, Sciences and Technology, 540142 Targu Mures, Romania; 8Close Circuit Pharmacy of County Clinical Hospital of Targu Mures, 540072 Targu Mures, Romania; 9Department of Pathology, Akdeniz University School of Medicine, Antalya Pınarbaşı, Konyaaltı, Antalya 07070, Turkey; 10Department of Cardiology, “Victor Babes” University of Medicine and Pharmacy, 2 Eftimie Murgu Sq., 300041 Timisoara, Romania; 11Department of Obstetrics and Gynecology, “George Emil Palade” University of Medicine, Pharmacy, Sciences and Technology, 540142 Targu Mures, Romania; 12Department of Medical Informatics and Biostatistics, “Victor Babes” University of Medicine and Pharmacy, 2 Eftimie Murgu Sq., 300041 Timisoara, Romania; 13Department of Cardiology, “Pius Brinzeu” County Hospital, 2 Eftimie Murgu Sq, 300041 Timisoara, Romania; 14Department of Pathophysiology, “George Emil Palade” University of Medicine, Pharmacy, Science, and Technology of Targu Mures, 540142 Targu Mures, Romania

**Keywords:** dysgenetic gonads, hermaphroditism, dysgerminoma, seminoma, oophorectomy, egg donation, embryo transfer

## Abstract

Ovarian malignant germ cell tumors (OMGCTs) represent a rare type of malignant tumors composed of primitive germ cells that often originate from dysgenetic gonads and are frequently associated with hermaphroditism. Such tumors occur more frequently in adolescents or young adults, and their etiopathogenic mechanism is not well established. We report the case of a 20-year-old female with ovarian dysgenesis and female phenotype. A laparoscopic surgery was performed, and ovotestis was discovered. To achieve a histopathological examination, right oophorectomy was performed, which confirmed the diagnosis of dysgerminoma. In the case of hermaphroditism, mixed germ cell tumors can develop, leading to a more aggressive evolution with bilateral malignancy of the gonads, which requires the removal of both ovotestis. The patient was recalled. A histopathological examination revealed a seminoma, so laparoscopic left oophorectomy was performed. The management of this type of diagnosis primarily involves surgery, minimally invasive interventions being preferred. Not all pathologic conditions are readily identifiable by means of exploratory laparoscopy, as in our patient’s case. We consider that the optimal solution for these patients would be the preservation of fertility via egg donation and embryo transfer; the survival rate of such patients being 97–100%, if the tumor is diagnosed at an early age.

## 1. Introduction

Malignant ovarian germ cell tumor is a rare tumor composed of primitive germ cells frequently associated with hermaphroditism [1,2,3]. Hermaphroditism was first described by Swyer in 1955 in a patient with female external genitalia [4]. About 75% of these tumors are discovered in stage I according to International Federation of Obstetrics and Gynecology (FIGO) staging [1,5]. They are usually unilateral, but they can also develop bilaterally, usually unaccompanied by ascites [1].

Frequently associated with dysgenetic gonads and hermaphroditism [6], this type of tumor is more common in young adulthood. Ovotestis or OT-DSD (ovotestis disorder in sex development) formerly known as true hermaphroditism is a pathology in which the individual has both ovarian and testicular tissue, irrespective of patients karyotype, with an incidence of 1/100,000 live births [4,6,7]. Ovotestis-DSD is a rare condition, accounts for only 3–10% of DSD, not all DSD is ovotestis [4,6,8]. OT-DSD is associated with abnormal development of internal and external genital organs and heterogenous presentation [8].

More studies are needed to find out the etiopathogenic mechanism because it is still unclear [4]. Possible causes may be any mutation of SRY gene, karyotype 46xy, mosaicism 46xx/46xy exposure to exogenous sex hormones [6]. Dysgerminoma occurs at a fertile age; the most prominent characteristic tumor markers of dysgerminoma are HCG (human chorionic gonadotropin) and LDH (lactate dehydrogenase) to help to diagnose this tumor [1,2,3]. Tumors that develop from dysgenetic gonads have a much more aggressive evolution with bilateral malignancy of the gonads. Ovarian dysgerminoma are rare malignant tumors; they are female analogous to male seminoma [2]. In this case, the removal of both ovotestis is the indicated management and minimally invasive interventions are preferred [1,9]. Laparoscopy does not always detect pathological lesions. Postoperatively, adjuvant chemotherapy can improve the prognosis, even if it is not always indicated [2,10]. In terms of the diagnosis and treatment of these tumors, a standardization was attempted with multidisciplinary participation under the guidance of the Malignant Germ Cell International Consortium (MaGIC) founded in 2009, which collaborates in this regard with the European Society of Gynecology Oncology (ESGO) and the European Society for Paediatric Oncology (SIOPE) [11].

After bilateral oophorectomy, hormone replacement therapy is required. Preservation of fertility with egg donation and embryo transfer is the optimal solution because the survival rate of such patients is 97–100% if the tumor is diagnosed at an early stage [10,12,13].

## 2. Case Presentation Section

A 20-year-old phenotypically female patient with no significant symptoms, only a secondary amenorrhea, presented to our hospital with the suspicion of gonadal dysgenesis. Family histology was irrelevant, no history of genitourinary anomalies, or infertility, or genetic disease. Physical examination revealed normal female external genital organs. An ultrasound examination revealed uterine hypoplasia, and no ovaries were detected.

On 4 October 2021, an exploratory laparoscopy was performed on the patient, confirming a hypoplastic uterus. Instead of ovaries, streaks of connective tissue were identified. Subsequently, adhesiolysis and right oophorectomy were performed. The patient’s postoperative evolution under antibiotic, anticoagulant, antianalgesic, and anti-inflammatory treatment was favorable. At the pathological examination, the microscopy and immunohistochemical evaluation were consistent with the diagnosis of dysgerminoma (Figure 1A–C). Histologic features of dysgerminoma were found, characterized by nests and nodules of uniform tumor cells, separated by fine connective tissue, containing inflammatory cells; the tumor cells were polygonal in shape positive for hematoxylin eosin staining, along with ovarian tissue with multiple primary follicles. Immunohistochemical staining positive for c-kit/cluster of differentiation 117 (CD 117) and immunohistochemical staining positive for placental alkaline phosphatase (PLAP). Considering the intraoperative aspect of the tumor suggestive for ovotestis, we highly suspected hermaphroditism and due to the high risk of malignancy of the remaining gonad decided to recall the patient in order to complete the surgical treatment by bilateral oophorectomy. The evolution was favorable, and the patient was discharged. The surgical specimen was sent to the pathology department for histological and immunohistochemical evaluation. The microscopy was consistent with the diagnosis of testicular seminoma, the equivalent of ovarian dysgerminoma in the testicle, a tumor with reduced malignancy and good prognosis after treatment. After the pathological examination of the remaining gonad, the microscopy was found to be consistent with the diagnosis of germ-cell neoplasia in situ (GCNIS) (Figure 1D,E, black square in Figure 1H, and Figure 1J), associated with testicular seminoma (Figure 1F,G, black arrow in Figure 1H, Figure 1I,K), represented by seminiferous tubules, characterized by large germ cells with vacuolated cytoplasm, with enlarged, hyperchromic, and pleomorphic nuclei located along the base membrane, interrupted by fibrous septa, Sertolli cells. (E) Details of the area shown in panel D; hematoxylin and eosin stain. Immunohistochemical staining positive for c-kit/cluster of differentiation 117 (CD 117) and immunohistochemical staining positive for placental alkaline phosphatase (PLAP). It is the equivalent of ovarian dysgerminoma in the testicle, also a tumor with reduced malignancy and good prognosis after treatment. Combining the clinical examination with the histological characteristics, the diagnosis of dysgerminoma and seminoma in OT-SDS was confirmed.

To establish the complete diagnosis of hermaphroditism, the patient was asked to undergo karyotype determination. Subsequently, a follow-up for beta-human chorionic gonadotropin (β-HCG), lactate dehydrogenase (LDH), carcinoembryonic antigen (CEA), and α-fetoprotein biomarkers was recommended. The patient was sent to the Oncology Department. Even if, according to the World Health Organization (WHO), the tumoral stage was clinically, surgically, and histologically IA, we could not exclude the need for chemotherapy treatments with platinum derivatives such as cisplatin, etoposide, and bleomycin. Hormone replacement therapy was also required. In this situation, fertility can be preserved through oocyte donation and embryo transfer. Unfortunately, we cannot further report any specific data about the patient because she did not return for follow-up.

## 3. Discussion

Ovarian malignant germ cell tumors (OMGCTs) represent a rare type of malignant tumors composed of primitive germ cells that often originate from dysgenetic gonads and are frequently associated with hermaphroditism. Such tumors occur more frequently in adolescents or young adults, and their etiopathogenic mechanism is not well established [1,9]. Ovotestis refers to the presence of both ovarian and testicular tissue in the same individual, and it represents 5% of sex development disorders, regardless of the patient’s karyotype. It is extremely rare, and its prevalence rate is considered less than 1/20,000 [6]. The risk of cancer increases in ovotestis, and its karyotype can be 60% 46XX, 33% mosaicism 46XX/46XY, and only 7% 46XY [4,6,14,15]. The causality of dysgenesis is not precisely known. One explanation may be abnormalities of the sex chromosomes, but gonadal and endocrine abnormalities during embryonic development may also play a role in its pathogenesis. The SRY gene may be damaged, but the presence of the testicle is less common [5,16]. Dysgerminomas represent 1–2% of the ovarian neoplasia, the same percent as seminomas, their analogue germ cell tumors [1,2,3].

Regarding intraoperative staging, unless peritoneal washing is involved, lymphadenectomy or omentectomy is not recommended, these being used in advanced stages II or III [5,11]. There is also no tumor grading system in the case of germinal tumors, with the exception of teratomas [5]. According to the literature, the prognosis is favorable after surgery, with or without chemotherapy [2,10]. In these tumors, a poor prognosis is mainly due to their association with other types of germinal tumors, which complicates the staging, an example being the association of these tumors with yolk sac tumors [5,11]. Fertility preservation involves some assisted reproduction techniques and hormone replacement therapy [10,12,13].

A second-look surgery is useful only if the biomarker levels increase [5]. In the case of tumor recurrence, which occurs in 20% of the cases, platinum-based chemotherapy is preferred, radiotherapy not being a common option [2,10,11,17,18]. The most frequent adverse effects of chemotherapy are fever and thrombocytopenia [19].

There is no general consensus, but a follow-up is recommended every 3 months in the first 3 years and every 6 months for the next 2 years. In the next 10 years, an annual evaluation should be sufficient [20]. Follow-up consists of clinical examination; imaging; identification of biomarkers, especially beta-human chorionic gonadotropin (β-HCG); and chest X-ray for metastases [1,18].

FIGO guidelines recommend surgical staging, but ESGO and SIOPE do not [21,22]. ESGO and SIOPE do not even recommend chemotherapy in children, due to its adverse effects. Therefore, the treatment of these cases represents a challenge, requiring multidisciplinary teams at reference centers trying to standardize management [11,23].

However, the treatment of these tumors is the most successful, with patients responding well to the surgical removal of the tumors and showing a survival rate of 60–80% in advanced stages and even 100% in early stages [11,19,24]. Chemotherapy increases survival, even in advanced stages, by up to 98% [11]. If a patient shows resistance to chemotherapy, immunotherapy or stem cell transplantation is attempted [11,25].

Panel and genetic sequencing would also be useful; fewer than ten cases of tumors arising in the gonads of individual with 46XX OT-DSD have been reported, mostly seminoma and frequently in male with this karyotype. Tumor samples can be used to identify mutations. This technique can be used to provide the genetic features of the rare tumor MOGCTs type in the ovary [7,26].

The strong point of the presented case is the rarity of this histological association, dysgerminoma on one of the gonads and seminoma on the contralateral one made tumor staging a real challenge. Dysgerminomas represent 1–2% of the ovarian neoplasia, the same percent as seminomas, their analogue germ cell tumor [1,2,3]. However, this association between dysgerminoma and seminoma in the same individual is extremely rare, that is why it is so important to report such an association [6]. In addition, in this patient, the tumor was discovered precociously, before the tumoral mass became evident by imaging or even by laparoscopy. The weak points are represented by the lack of a karyotype genetic test, which would have helped make a complete diagnosis of hermaphroditism; the lack of an oncological follow-up; and no evidence of required biomarkers recommended by the literature [1,12,20,27] for further management. However, the survival rate is known to be high after this type of therapy, and surgery is considered the gold standard.

Considering our patient’s age, her fertility can be preserved using assisted human reproduction methods, which include egg donation and embryo transfer, provided she undergoes hormone replacement therapy, which is mandatory in her case.

The treatment of this type of tumors represents one of the greatest successes in gynecological oncology; the survival rate of the patients increasing even without lymphadenectomy or omentectomy, which can be performed during surgical staging, if required, along with therapy in the more advanced stages. Platinum-based chemotherapy remains a controversial topic in pediatric treatment. However, in adults, it has proven its usefulness in the long-term prognosis. At reference centers, multidisciplinarity and therapy should be the conduct to follow. The management of OT-DSD requires a multidisciplinary approach because of its specific condition [8]. Gender dysphoria can occur and the reason why it is essential to involve the patient in the therapeutic decision and respect their autonomy. Follow-up is necessary to include a physical examination, hormonal evaluation, genetical panels and electrolyte analyses to avoid adrenal failure [8]. Cases of OT-DSD must be treated based on their external genitalia and the orientation of the patient [6].

## 4. Conclusions

This association of dysgerminoma with seminoma is rare, but the prognosis is favorable. The diagnosis is exclusively histopathological, and the gold standard in management remains laparoscopy with extirpation of the gonads. Fertility preservation is one of the basic concerns in the case of young people; in this case, it is possible through egg donation and embryo transfer, associated with hormone replacement therapy.

## Figures and Tables

**Figure 1 reports-06-00014-f001:**
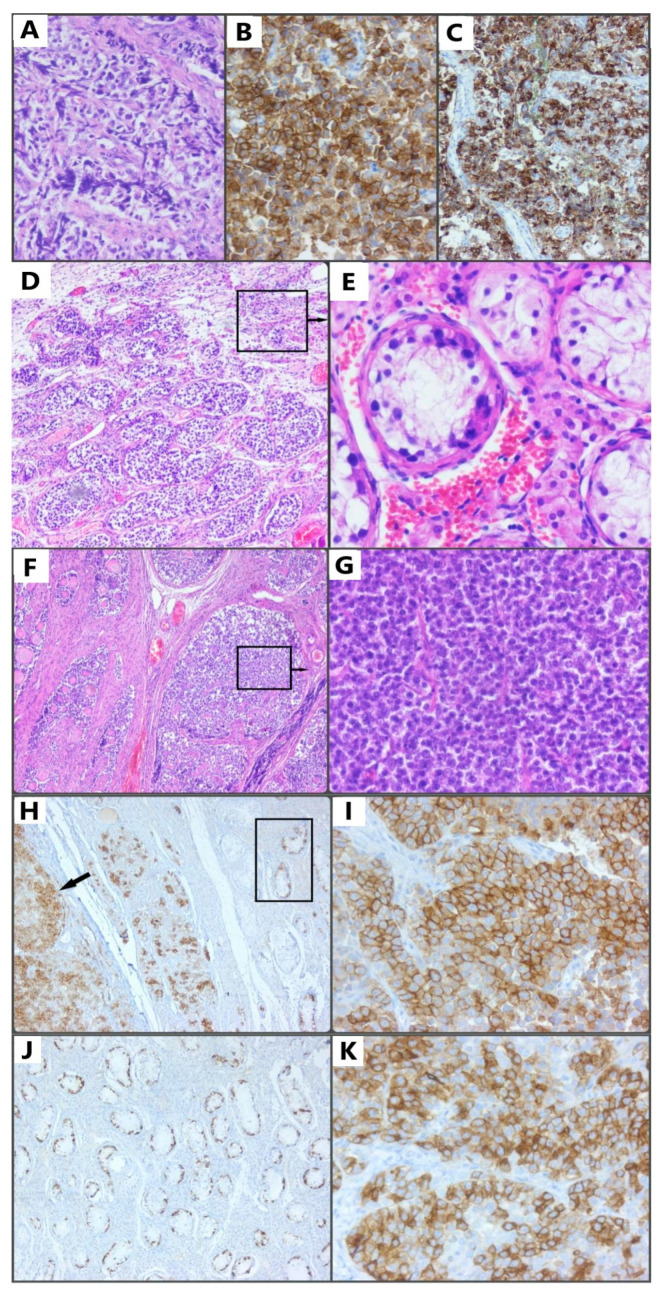
(**A**) Histologic features of dysgerminoma characterized by nests and nodules of uniform tumor cells, separated by fine connective tissue, containing inflammatory cells; the tumor cells are polygonal in shape, with clearly visible cell borders, an eosinophilic-to-clear cytoplasm, a round nucleus, and centrally located and prominent nucleoli; hematoxylin and eosin stain (HE, ob. 20×). (**B**) Immunohistochemical staining positive for c-kit/cluster of differentiation 117 (CD 117); original magnification 10×. (**C**) Immunohistochemical staining positive for placental alkaline phosphatase (PLAP); original magnification 20×. (**D**) Germ cell neoplasia in situ (GCNIS) represented by seminiferous tubules, characterized by large germ cells with vacuolated cytoplasm, with enlarged, hyperchromic, and pleomorphic nuclei located along the base membrane, interrupted by fibrous septa. (**E**) Details of the area shown in panel D; hematoxylin and eosin stain (HE, ob. 10× and 20×). (**F**) Testicular seminoma consisting of a scattered tumoral pattern, interrupted by fibrous septa, with uniform tumoral cells represented by a clear-to-eosinophilic cytoplasm and round or oval nuclei, with finely granular chromatin and flattened edges. (**G**) Details of the area shown in panel F; hematoxylin and eosin stain (HE, ob. 10× and 20×). (**H**) Immunohistochemical staining positive for c-kit/cluster of differentiation 117 (CD 117); the black arrow indicates testicular seminoma, and the black square highlights germ cell neoplasia in situ (GCNIS). (**I**) Details of testicular seminoma, the infiltrative component. (**J**) Immunohistochemical staining positive for placental alkaline phosphatase (PLAP): germ cell neoplasia in situ (GCNIS); original magnification 10×. (**K**) Immunohistochemical staining positive for placental alkaline phosphatase (PLAP): testicular seminoma; original magnification 20×.

## Data Availability

Department of Pathology, County Clinical Hospital of Targu Mures, 540072 Targu Mures, Romania; Department of Obstetrics and Gynecology, “George Emil Palade” University of Medicine, Pharmacy, Sciences and Technology.

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
