# Peer review of "A Rare Case of Malignant Ovarian Germ Cell Tumor: Dysgerminoma and Seminoma in the Same Patient"

_reports, 2023, doi:10.3390/reports6010014_

Round 1

Reviewer 1 Report

A case report of a phenotypical female is presented with dysgentic gonads in which bilaterally a dysgerminoma is diagnosed. As far as I know many details are incorrect and after correction a case of bilateral dysgerminoma in a phenotypical female with the diagnosis of disorder in sex development (DSD) patient is described. This is not very interesting to report. A few examples of the inaccuracies:

-      As far as I'm aware a dysgerminoma and a seminoma are the same tumor (depending on the gonad) and typical in a dysgenetic gonad in a 46xx individual (not being a typical ovary or testis).

-      I would be very interested in the further workout of this patient as to the genotype and the external genitalia.

-      The term hermaphroditism should be replaced by "disorder of sex development".

-      The incidence of DSD in dysgerminoma is described as high, but to my knowledge around 5%, whereas the incidence of dysgerminoma in 46XX DSD is much higher.

The authors could rewrite the case report with the correct details but lacking the workout of the DSD makes it less interesting.

Author Response

Dear reviewer

In this case report we describe a dysgerminoma and a seminoma in disgenetic gonads, testicular seminoma  is counterpart for ovarian dysgerminoma, but they are not exactly the same. Histologic features of dysgerminoma, characterized by nests and nodules of uniform tumor cells, separated by fine connective tissue, containing inflammatory cells; the tumor cells are polygonal in shape, along with ovarian tissue with multiple primary follicles. Dysgenetic testicular cords with Sertoli cells, cells sugestive of testicular seminoma  consisting of a scattered tumoral pattern, interrupted by fibrous septa, with uniform tumoral cells represented by a clear to eosinophilic cytoplasm and round to oval nuclei, with finely granular chromatin and flattened edges .Line 108-114, 127-132

Unfortunately she did not return for genetic testing .Physical examination revealed normal female external genital organs.Line 98.

Ovotestis or OT-DSD formarly known as true hermaphroditism is a pathology in which the individual have both ovarian and testicular tissue, irrespective of patients karyotype,with an incidence of 1/100,000 live birth.[4,6,7]line 67-70

 Ovotestis-DSD is a rare condition, accounts for only 3-10% of DSD, not all of DSD is ovotestis. [4,6,8] OT-DSD is associated with abnormal development of interna land external genital organs and heterogenous presentation.[8]line 70-73

Dysgerminomas represent 1-2% of the ovarian neoplasia, the same procent as seminomas, their analogue germ cell tumor [1,2,3]. But this association between dysgerminoma and seminoma in the same individual is extremly rare, that is why is so important to report such an association.[6] line 274-2767

The management of OT-DSD requires multidiciplinary approach because of its specific condition[8]Gender disphoria can occur, reason why it is essential to involve the patient in the therapeutic decision and respect his autonomy. Follow-up is necessary toinclude physical examination, hormonal evaluation, genetical panels, electrolyte analyses to avoid adrenal failure.[8].Cases of of OT-DSD must be treated based on their external genitalia and the orientation of the patient.[6] line297-303

Thank you very much!

Reviewer 2 Report

The authors present a case of dysgerminoma on one of the gonads, and seminoma on the contralateral one.

-       General comment:  English language is not adequate and should be revised. 

-       What is your protocol for prophylactic anticoagulants and antibiotics after laparoscopic procedures

-       The conclusion is too long. You should point out key aspects of this case report.

Author Response

Dear reviewer

English language was revised by MDPI editing service.

Our antibiotic prophylaxis protocol consists of an immediate postoperative dose in case of laparoscopy.The prophylactic anticoagulant is administeredat 12 hours after laparoscopy, one single dose.

I kept only the important ideas of the conclusion, the rest being replaced to the discussion.

Thank you very much

Reviewer 3 Report

Major issues:

The most prominent characteristic tumor markers of dysgerminoma are hCG and LDH to help diagnose dysgerminoma. You should include those information in introduction part.

Testicular seminoma is more prevelant and it is a counterpart to ovarian dysgerminoma. You can add those information to your introduction part.

In discussion part, sequencing technique may be used to help find the targets for this rare disease. Genomic landscapes of it can also be characterized.

Author Response

Dear reviewer

The most prominent characteristic tumor markers of dysgerminoma are HCG [ human chorionic gonadotropin] and LDH [ lactate dehydrogenase]to help to diagnose this tumor.I introduced this information in the introduction line 77-79.

 Ovarian dysgerminomas are rare malignant tumor, they are female analogous to male seminoma.[2]line80-81.

 Panel and genetic sequencing would also be useful, less than ten cases of tumors arising in the gonads of individual with 46xx OT-DSD have been reported , mostly seminoma and frequently in male with this karyotype. Tumor samples can be used to identify mutations .This technique can be used to provide the genetic features of the rare tumor MOGCTs type in the ovary.[7,26].line 267-270

Thank you

Round 2

Reviewer 2 Report

No further comments